# PARTITIONING-GUIDED K-MEANS: EXTREME EMPTY CLUSTER RESOLUTION FOR EXTREME LANGUAGE MODEL COMPRESSION

## ABSTRACT

Compactness in deep learning can be critical to a model's viability in low-resource applications, and a common approach to extreme model compression is quantization. We consider Iterative Product Quantization (iPQ) with Quant-Noise (Fan et al., 2020) to be state-of-the-art in this area, but this quantization framework suffers from preventable inference quality degradation due to prevalent empty clusters in language modeling tasks. In this paper, we propose several novel enhancements aiming to improve the accuracy of iPQ with Quant-Noise by focusing on resolving empty clusters. Our contribution, which we call Partitioning-Guided k-means (PG k-means), is a heavily augmented k-means implementation composed of three main components. First, we propose a partitioning-based pre-assignment strategy that minimizes initial empty clusters and encourages an even weight-to-cluster distribution. Second, we propose an empirically superior empty cluster resolution heuristic executed via cautious partitioning of large clusters. Finally, we construct an optional optimization step that consolidates intuitively dense clusters of weights to ensure shared representation. The proposed approach consistently reduces the number of empty clusters in iPQ with Quant-Noise by 100x on average, uses 8x fewer iterations during empty cluster resolution, and improves overall model accuracy by up to 12%, when applied to RoBERTa on a variety of tasks in the GLUE benchmark.

## 1 INTRODUCTION

There is a more critical need than ever for compact, but effective, deep learning models in an age where even minimal models may have hundreds of millions of parameters. With the recently explosive popularity of truly large language models (LLMs), achieved primarily through scaling compute resources, the constraints of low-resource deployment environments must be freshly considered and addressed. Given that, effective model compression is a research area of significant interest. A number of simple and popular compression methodologies exist, such as weight sharing (Dehghani et al., 2018), weight pruning (LeCun et al., 1989), or knowledge distillation via teacher-student relationships during training (Hinton et al., 2014; Sanh et al., 2019; Jiao et al., 2019), but these are most applicable for models that are over-parameterized.

Quantization is an alternative approach, and it reduces the memory footprint of weights for a model by generally reducing the number of bits per weight for that weight's representation. Various quantization methodologies exist (Gupta et al., 2015; Courbariaux et al., 2015; Stock et al., 2020), but Iterative Product Quantization (iPQ) with Quant-Noise (Fan et al., 2020) enabled during training and/or fine-tuning has cemented itself as the state-of-the-art for quantization. iPQ with Quant-Noise improves on the performance of several competitive predecessors (Stock et al., 2020; Jacob et al., 2017) for extreme compression (referring to compression ratios of 10x or more), but issues still remain.

A notable problem for many quantization methods is empty cluster resolution, which is ultimately a NP-hard problem for modern clustering algorithms. We posit that the presence of empty clusters often leads to noteworthy losses in inference quality, so we consider their minimization a priority. Generally, we find that iPQ with Quant-Noise suffers from a significant number of unresolved empty clusters (e.g., over a hundred empty clusters for a linear layer; more details later) and that there

is considerable performance degradation associated with this (e.g., observing a 2.7% difference in accuracy between models featuring an empty cluster resolution heuristic and models without one). In this paper, we start with going over the empty cluster problem in detail, analyzing the number and distribution of empty clusters across compression ratios and layers for models quantized with iPQ with Quant-Noise, and providing a brief, intuitive explanation as to how empty clusters lead to performance degradation.

To better address the empty cluster problem for extreme model compression, we propose *Partitioning-Guided k-means (PG k-means)*, which is composed of several novel and effective techniques to improve the clustering algorithm typically employed by iPQ with Quant-Noise in extreme compression applications. The proposed scheme includes three major contributions. First, we propose a replacement for the typically random (or influenced random) placement of initial centroids with a pre-assignment strategy that minimizes initial empty clusters and guides k-means towards a roughly even distribution of weight assignments to clusters. Second, we propose an empirically superior empty cluster resolution heuristic executed via cautious partitioning of populous clusters into new sub-clusters. Finally, we construct an optional optimization step that consolidates dense clusters of weights to ensure that they map to a single centroid after quantization completes and are not erroneously/unintentionally separated.

To validate the viability of this approach, we test our complete method on RoBERTa Liu et al. (2019) fine-tuned for several tasks in the GLUE benchmark. When compared directly to the state-of-the-art in iPQ with Quant-Noise, our method reduces the average number of empty clusters on a layer-by-layer basis by 100x on average, reduces the number of layers with empty clusters consistently by at least 25x, and typically undergoes 8x fewer iterations for empty cluster resolution. Moreover, the proposed PG k-means consistently supersedes the accuracy scores of iPQ with Quant-Noise by up to 2.4% for MNLI, up to 12% for RTE, and up to 4.2% for QNLI, all on extremely compressed models.

## 2 BACKGROUND

We focus our brief review of existing literature on popular methods of quantization with a focus on extreme compression. Weight-sharing (Dehghani et al., 2018), weight-pruning (LeCun et al., 1989), and knowledge distillation (Hinton et al., 2014; Sanh et al., 2019; Jiao et al., 2019) are useful compression methods, but are not our focus and are synergistic to our method. Fixed-point scalar quantization (Gupta et al., 2015; Courbariaux et al., 2015) is also a popular quantization method, but tends to be unsuitable for high compression ratios when employed alone, and as such is not covered here.

### 2.1 POPULAR QUANTIZATION METHODOLOGIES

Product quantization (PQ) is a long-time solution for extreme compression applications. PQ is a subset of the more general form of vector quantization (VQ) that, for a given set of weights in a matrix for a layer $W_l$, learns a codebook filled with code-words for each column of that weight matrix. Compression with PQ is accomplished via the division of each column of $W_l$ into some $m$ vectors per column $c$, with $m \times c$ total vectors. All of these vectors share the same layer-wide codebook instead of one per column. Codebooks are typically determined via several iterations of a classical k-means algorithm (Lloyd, 1957) with a fixed number of $k$ centroids such that the reconstruction error is minimized, although this is customizable to any clustering algorithm.

Iterative product quantization (iPQ) was proposed by Stock et al. 2020 to minimize the significant performance degradation that often occurs in vanilla PQ in two ways: by focusing on minimizing the error of the reconstructed output of a given layer as opposed to the reconstructed weights and by doing so in an iterative manner from layer to layer. Intuitively, quantizing online while training or fine-tuning and layer-by-layer allows later layers to adjust as they examine the quantized output of previous layers, conditioning reconstruction error robustness. iPQ remains a state-of-the-art quantization method for generalizable extreme compression, although enhancements have been proposed (Fan et al., 2020).

Table 1: Average number of empty clusters (lower is better) per layer type in RoBERTa quantized with typical iPQ with Quant-Noise and fine-tuned for MNLI, RTE, and QNLI. All results are derived from quantized models with compression ratios of 11.81 (left) and 15.9 (right). The total number of clusters for linear layers was 3072 and for embedding layers was 768.

| Compression Ratio of 11.81 | | | | Compression Ratio of 15.9 | | | |
|---|---|---|---|---|---|---|---|
| **Layer Type** | **MNLI** | **RTE** | **QNLI** | **Layer Type** | **MNLI** | **RTE** | **QNLI** |
| Embedding | 0.0 | 0.0 | 0.0 | Embedding | 0.0 | 0.0 | 0.0 |
| q_proj | 28.5 | 31.5 | 32.3 | q_proj | 121.7 | 114.2 | 122.1 |
| k_proj | 30.6 | 30.5 | 30.3 | k_proj | 119.3 | 119.0 | 115.3 |
| v_proj | 25.8 | 28.8 | 27.5 | v_proj | 108.3 | 111.2 | 114.8 |
| out_proj | 28.6 | 27.7 | 26.4 | out_proj | 89.1 | 95.2 | 93.1 |
| FC1 | 6.4 | 6.2 | 6.0 | FC1 | 6.9 | 8.3 | 7.4 |
| FC2 | 4.8 | 4.2 | 4.9 | FC2 | 0.1 | 0.3 | 0.0 |

## 2.2 QUANTIZATION AWARE TRAINING AND QUANT-NOISE

Expanding on these previous methods, Fan et al. focus on their application during training, ensuring that challenges such as null gradients during backward passes for quantized weights and widespread drift in network output are met with capable solutions. Straight-through estimators (STEs) are commonly used to deal with gradient issues for Quantization Aware Training (QAT) (Jacob et al., 2017; Bengio et al., 2013; Courbariaux & Bengio, 2016), but significant bias can still be introduced. In response, Quant-Noise (Fan et al., 2020) is proposed as a methodology that quantizes only a randomly selected portion of the weights of a given layer during training and fine-tuning, mitigating the bias introduced by STEs and still conditioning the network for reconstruction error robustness. iPQ with Quant-Noise during training and fine-tuning forms the current state-of-the-art for highly generalizable and extreme model compression.

## 3 EMPTY CLUSTERS ISSUE IN EXTREME MODEL COMPRESSION

### 3.1 HEURISTICS FOR EMPTY CLUSTER RESOLUTION

Empty clusters are a classical problem in k-means algorithms. Depending on the application, unresolved empty clusters can be numerous and may cause considerable performance loss. Most k-means implementations host some empty cluster resolution heuristics to mitigate the number of degenerate solutions (Aloise et al., 2017; Torrente & Romo, 2020; Chun, 2021; Feiping et al., 2022). However, there is no theoretical guarantee that all empty clusters are resolved within reasonable run-time and these heuristics are not always widely applicable. Fairseq's (Ott et al., 2019) iPQ with Quant-Noise implementation hosts a computationally efficient mixture of two popular heuristics, $\epsilon$-greedy and $\epsilon$-random (Aloise et al., 2017). Upon encountering an empty cluster, their mixed strategy greedily chooses the most populous non-empty cluster, bases a new centroid off of the one of the populous cluster, and randomly perturbs both.

### 3.2 INCREASED EMPTY CLUSTER OCCURRENCE IN EXTREME MODEL COMPRESSION

While efficient, we find that the popular empty cluster resolution heuristic employed by iPQ with Quant-Noise struggles to completely resolve empty clusters for quantized RoBERTa models fine-tuned for tasks on the GLUE benchmark, and the issue generally aggravates when the model is compressed more. Table 1 demonstrates the average number of empty clusters per type of layer produced by iPQ with Quant-Noise on various tasks within the GLUE benchmark for compression ratios of 11.81 and 15.9. We note that for many layer types, deeper quantization tends to produce more empty clusters, aligning with inference quality degradation for deeper compression ratios. Clearly, empty clusters are prevalent and need to be addressed for extreme model compression.

### 3.3 QUALITY DEGRADATION FROM EMPTY CLUSTERS IN MODEL QUANTIZATION

Loss of prediction quality is often observed in the presence of empty clusters. Part of this is due to a corresponding loss in model expressivity. For a layer in a poorly quantized model with dozens of empty clusters, its range of outputs is artificially limited. As a trivial example, if those dozens of empty clusters were to be filled with just a single weight each such that the centroids of those clusters corresponded directly to each weight, the expressivity of the layer necessarily improves (assuming non-trivial weight distributions). Given that, the presence of empty clusters is necessarily sub-optimal and their minimization should be a priority, although heuristics that attempt to resolve empty clusters need to be cautious to avoid drifting from locally optimal solutions. In practice, we find that for iPQ with Quant-Noise, a significant loss in quality occurs when no empty cluster resolution heuristic is applied for quantizing RoBERTa fine-tuned for MNLI, producing a model with an accuracy of 76.2% versus a model with an accuracy 79.0% with the mixed heuristic this baseline natively employs.

### 3.4 EFFECTS OF CODEBOOK PRUNING FOR EMPTY CLUSTERS

It is worth noting that a natural counterpoint to the issues with empty clusters would be to propose pruning of the PQ codebook for those useless centroids to improve a given quantized model's compression ratio. While this can be done, in practice, we found that for most applications this would only improve the compression ratio by less than one percent (e.g. a compression ratio of 15.29 would shift to 15.31 for MNLI results for iPQ with Quant-Noise). Given that, we do not consider this moving forward for our tests. If empty cluster pruning would have a significant effect on the compression ratio of a model, it is likely that the model is poorly quantized to begin with and its performance for that compression ratio would be compromised.

## 4 PROPOSED: PARTITIONING-GUIDED K-MEANS (PG K-MEANS)

To better address problems associated with empty clusters and improve overall prediction quality, we propose *Partitioning-Guided k-means (PG k-means)*, a novel k-means implementation loosely inspired by binary-space partitioning applied towards an empirically superior pre-assignment strategy and empty cluster resolution. Our scheme focuses on encouraging an initially even distribution of weights to clusters and guarantees zero empty clusters for the initial state of k-means. Additionally, our method seeks to resolve empty clusters during k-means iterations by splitting up populous clusters into new, smaller sub-clusters. While our method does not provide theoretical guarantees for reducing the number of empty clusters, in all target applications our tests showed a minimized number of empty clusters when compared to the state-of-the-art iPQ with Quant-Noise, and for many applications all empty clusters were resolved. Our proposed algorithm, PG k-means, consists of three primary steps that heavily augment a typical k-means implementation: Partitioning-Guided Pre-assignment, Partitioning-Guided Cluster Fine-tuning, and an optional optimization called Dense Weights Consolidation. Detailed pseudo-code for PG k-means can be found in our supplementary materials.

### 4.1 PARTITIONING-GUIDED PRE-ASSIGNMENT

The performance of k-means implementations depends heavily on the pre-assignment strategy defining the initial placement of centroids. While random placement, or influenced random placement, is somewhat popular and is employed for k-means in iPQ with Quant-Noise, such strategies can result in significant variation in final cluster assignments. Moreover, such pre-assignment strategies commonly lead to numerous empty clusters that need resolution. In response, we propose an alternative that we call *Partitioning-Guided Pre-assignment*.

Our pre-assignment strategy focuses on guaranteeing that no empty clusters are present initially for non-trivial weight distributions, without relying on an empty cluster resolution heuristic. Here, we use the term "weight distribution" to refer to the distribution of the weights (i.e., data points) that are being quantized in the $n$-dimensional space. In order to accomplish this, our method constructs initial clusters by recursively bisecting the overall weight distribution, guiding k-means towards roughly even assignments of weights to each cluster and minimizing initial empty clusters. Specifically, Partitioning-Guided Pre-assignment begins by assigning a temporary centroid for the entire set of

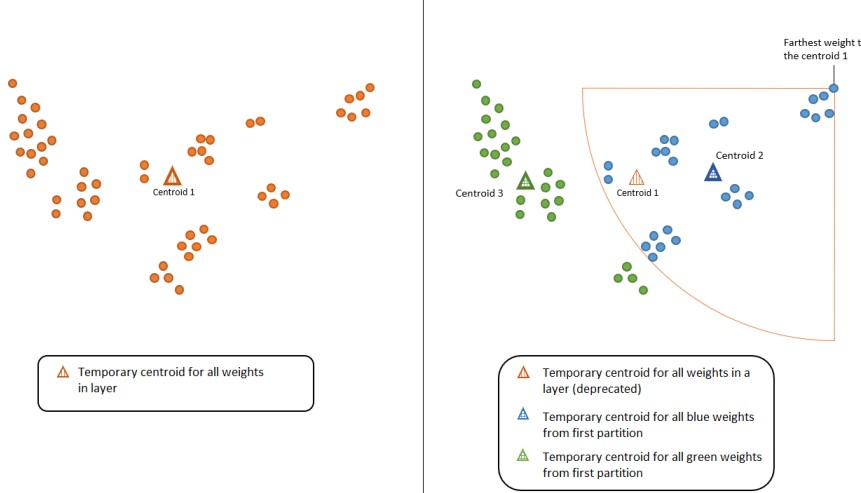

Figure 1: Illustration of Partitioning-Guided Pre-assignment across two partitioning time-steps when applied to a synthetic distribution. Tentative clustering is decided via $n$-dimensional, spherical partitions centered on the farthest point within the cluster of a given tentative centroid. The radius of the spherical partition targets a dynamically determined number of weights that would be assigned to the new clusters.

weights in a layer, labelled as "Centroid 1" in Figure 1. An $n$-dimensional sphere is then constructed to roughly bisect the overall weight distribution into two clusters. This sphere is centered on the weight that has the furthest Euclidean distance from the temporary centroid (e.g., top-right point in Figure 1), intuitively the data point with the worst representation in the temporary cluster. Upon the temporary cluster being bisected, the temporary centroid is removed and replaced by two new centroids that are generated for the two new clusters, corresponding to "Centroid 2" and "Centroid 3" in the figure. This strategy is executed recursively on the new clusters until the desired number of centroids have been determined.

While Partitioning-Guided Pre-assignment bisects temporary clusters at every time-step, we note that the method for determining the radius of the partitioning sphere is customizable. Our proposed method focuses on enforcing a roughly even distribution of assigned weights to clusters, but alternatives with different goals could improve performance. We leave it to future work to investigate the potential of these alternatives.

## 4.2 PARTITIONING-GUIDED CLUSTER FINE-TUNING

While a more even distribution of assignments via the execution of Partitioning-Guided Pre-assignment already minimizes the initial occurrence of empty clusters, they can still arise during k-means iterations. As k-means settles in a local optimum durings its iterations, the solution represented by that local optimum may call for fewer intuitive, or natural, clusters than prescribed at a high level. This produces a perceived overestimation of the number of clusters, where k-means can represent the same locally optimum solution with fewer centroids than are provided. However, as we have already covered, the presence of empty clusters is necessarily sub-optimal and their resolution is important to model performance. To enable extreme empty cluster resolution towards that end and seeking to push k-means out of these erroneous local optima, we propose *Partitioning-Guided Cluster Fine-tuning*.

At a high level, our method for empty cluster resolution seeks out populous clusters and attempts to split them into multiple smaller clusters. In order to split clusters efficiently, instead of bisecting each populous cluster until its size reaches the average cluster size of the entire weight distribution, we propose guiding splits by providing a target post-split cluster size that scales dynamically across iterations.

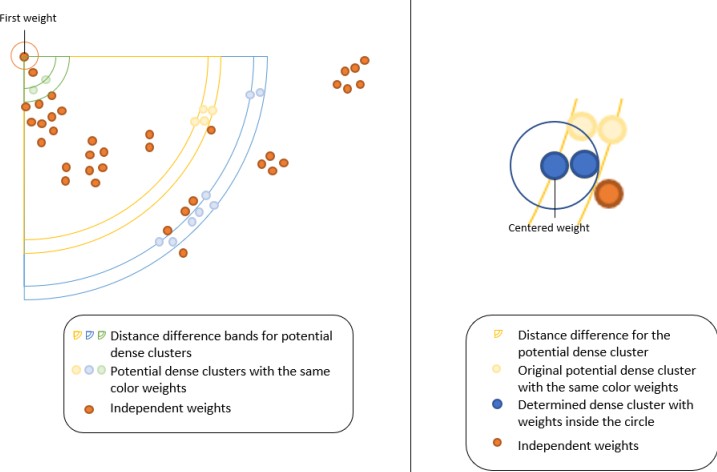

Figure 2: Illustration of Dense Weights Consolidation when applied to a synthetic distribution. Dense clusters are identified via a Euclidean distance-based criteria. Upon dense clusters being identified, they are replaced by a centroid representing that dense cluster and treated as a normal, singular weight for later clustering steps.

Intuitively, we could set the target cluster size simply as the average cluster size of all clusters larger than the layer-wide average. In practice, however, we have observed that this is too aggressive and can potentially split large, dense clusters into too many sub-clusters. Nevertheless, explicitly avoiding splitting dense clusters is difficult, as calculating the accurate cluster density can be computationally expensive. We propose a more efficient solution, detailed in Equation 1, that cautiously splits extremely large clusters by scaling the target cluster size alongside the size of the non-empty cluster. For Equation 1, we denote $n_{lc}$ as the number of weights in the non-empty cluster being split, $S_{avg}$ as the aforementioned adjusted average, and $S_{scl}$ as the scaling target cluster size. $\sqrt{n_{lc}/S_{avg}}$ is the number of small clusters that a large cluster would be split into assuming using $S_{avg}$ as the target, and the square root of that scales down the speed, preventing a large cluster from being partitioned into too many small clusters.

$$S_{scl} = max(\sqrt{n_{lc}}\sqrt{S_{avg}}, S_{avg}) \tag{1}$$

### 4.3 DENSE WEIGHTS CONSOLIDATION

This optional optimization is propelled by the observation that typical k-means and PG k-means without this augmentation will occasionally split up a dense cluster of weights such that those weights are mapped to separate, sometimes far-away, centroids. To address this issue, we propose *Dense Weights Consolidation* to ensure that a dense cluster, which should intuitively be represented by the same centroid, is preserved. To achieve that, assuming a dense cluster can be identified, we first use a single representative centroid to replace all the weights in the cluster. This representative centroid is used throughout later k-means iterations as if the cluster just has one weight. The cluster is mapped back to its original weights at the very end of k-means clustering.

A critical step in this optimization is to identify a dense cluster efficiently. We identify a dense cluster as a set of weights that fulfill two criteria. First, weights are identified as being potentially within a dense cluster, if the difference between their Euclidean distance to a randomly chosen anchor weight (e.g., the top-left weight in Figure 2 left) is less than a fine-tunable value $\varepsilon$. This corresponds to the rings of distance demonstrated in the figure. Second, the potential dense cluster is confirmed as a dense cluster if the distance between a random weight in that cluster to every other weight is less than $\varepsilon$, which corresponds to the dense weight confirmation via a centered weight observed in Figure 2 right. Perfectly determining sets of dense clusters is not feasible and is a subset of the well-studied NP-hard MIS problem. We propose our own heuristic to tackle this problem that performs well in our

experiments, striking a balance between computational efficiency and dense cluster identification quality.

The first step of our implementation chooses a random weight in our weight distribution as a focal point to construct a Euclidean distance map to every other weight. That distance map is subsequently sorted and iterated through to search for potential dense clusters, stopping whenever the difference between the distances of a set of weights fit our first established criteria. Upon establishing a set of weights that could form a dense cluster, that set is iterated through with an identified candidate weight $W_{cand}$. All other weights not fitting the first criteria are independent weights (i.e., not part of a dense cluster). For each potential dense cluster, the weights that fulfill the second identified criteria are paired with $W_{cand}$ and consolidated into a dense cluster and removed from the set of potential dense clusters. The rest of the weights in these potential dense clusters are considered independent weights and are not considered for other possible dense cluster sets. This process is repeated across the original distance map until all weights have been consolidated or classified as independent weights.

While $\varepsilon$ is a fine-tunable parameter, we found in our experiments that it was difficult to estimate good values of $\varepsilon$, and we suppose that ideal values for this parameter are likely layer-specific. Overestimation of $\varepsilon$, in particular, can cause degradation in quantization quality. In response, we propose scaling $\varepsilon$ dynamically to avoid over-identifying dense clusters. Equation 2 describes our update criteria, with $n_c$ corresponding to the number of centroids for the layer being quantized, $n_{cw}$ corresponding to the number of weights after consolidation, which is the sum of the number of dense clusters and independent weights, $c_{sd}$ corresponding to a scaling factor that reduces $\varepsilon$, $c_{mc}$ corresponding to the factor of multiple of $n_c$ that serve as a threshold for the minimum number of consolidated weights $n_{cw}$. $c_{sd}$ and $c_{mc}$ values of 0.8 and 2 respectively worked well in practice, indicating that if the number of weights after consolidation is less than twice the number of centroids, $\varepsilon$ is scaled by 0.8.

$$\varepsilon_{upd}(\varepsilon, n_c, n_{cw}, c_{sd}, c_{mc}) = \begin{cases} \varepsilon \times c_{sd} & \text{if } n_{cw} < n_c \times c_{mc}, \\ \varepsilon & \text{else} \end{cases} \qquad (2)$$

## 5 RESULTS

For our set of experiments, we employ Fairseq (Ott et al., 2019), a language and sequence modeling toolkit written in PyTorch that is fast, easily extendable, and hosts a Quant-Noise implementation. We make use of the provided Quant-Noise framework and Fairseq's iPQ implementation to apply our novel scheme to RoBERTa for several tasks within the GLUE benchmark. All cluster assignments were finished within 15 iterations of both respective k-means algorithms for each layer. During each k-means iteration, up to 100 iterations of typical iPQ with Quant-Noise's empty cluster resolution were allowed while up to 15 iterations of Partitioning-Guided Cluster Fine-tuning were allowed. Fine-tuning, quantization, and evaluation were performed on four NVIDIA Tesla V100s across all models.

### 5.1 PG K-MEANS FOR ROBERTA ON GLUE TASKS

All RoBERTa models were initially pre-trained checkpoints provided by Fairseq without quantization. These checkpoints were fine-tuned for MNLI, RTE, and QNLI tasks with Quant-Noise, using recommended noise factors between 0.05 and 0.2 and block sizes of 8. These baseline checkpoints were subsequently quantized either with typical iPQ or with our proposed method. Out of the available quantizable layers, we quantized the input embedding layer, all input and output projection layers related to encoder self-attention, and all fully connected layers, totaling to 73 layers overall. Exact quantization parameters can be found in our supplementary materials.

The results highlighted in Table 2 demonstrate a clear advantage for PG k-means compared to iPQ with Quant-Noise for MNLI, a task that was explored and used to validate the viability of iPQ with Quant-Noise. Concerning MNLI, our method demonstrates up to a 2.4% inference quality increase and consistently improves upon iPQ with Quant-Noise by at least 0.8% in the worst case. The difference between iPQ with Quant-Noise and our method grows for other tasks, with one example for RTE exhibiting a 12% accuracy increase from its iPQ with Quant-Noise baseline and QNLI demonstrating up to a 4.2% accuracy increase. Clearly, PG k-means consistently beats typical iPQ

Table 2: Complete validation set results of quantization implementations for RoBERTa fine-tuned for MNLI, RTE, and QNLI. The leftmost column contains compression ratios and the right columns contains accuracy scores in percentages. Best accuracy scores for a given compression ratio are bolded. The right table provides mappings between compression ratios to model size as a quick reference. All results were generated and are not reused from literature.

| RoBERTa base | | | |
| --- | --- | --- | --- |
| Compr. | MNLI | RTE | QNLI |
| Original Model | | | |
| 1.00 | **87.8** | **76.7** | **92.1** |
| iPQ with Quant-Noise | | | |
| 11.81 | 83.1 | 58.8 | 90.3 |
| 14.05 | 81.8 | 57.8 | 88.5 |
| 15.29 | 80.7 | 55.6 | 87.8 |
| 15.90 | 79.0 | 55.6 | 77.4 |
| PG k-means | | | |
| 11.81 | **83.9** | **70.8** | **90.5** |
| 14.05 | **83.3** | **59.6** | **88.9** |
| 15.29 | **82.0** | **56.7** | **87.9** |
| 15.90 | **81.4** | **56.3** | **81.6** |

| Compression Ratio | Size (MB) |
| --- | --- |
| 1.00 | 477.94 |
| 11.81 | 40.47 |
| 14.05 | 34.01 |
| 15.29 | 31.26 |
| 15.90 | 30.05 |

Table 3: Results for ablation study to demonstrate the isolated improvements of applying our optional Dense Weights Consolidation step to PG k-means to RoBERTa fine-tuned for MNLI. Best accuracy scores for a given compression ratio are bolded.

| Compr. | iPQ with Quant-Noise | Baseline PG k-means | Full PG k-means |
| --- | --- | --- | --- |
| 11.81 | 83.1 | 83.5 | **83.9** |
| 14.05 | 81.8 | 82.6 | **83.3** |
| 15.29 | 80.7 | 81.7 | **82.0** |
| 15.90 | 79.0 | 80.6 | **81.4** |

with Quant-Noise by a notable margin for several tasks in the GLUE benchmark when applied to RoBERTa, establishing its viability for extreme model quantization.

## 5.2 ABLATION STUDY OF PG k-MEANS ON MNLI

As PG k-means is composed of an optional optimization in the form of Dense Weights Consolidation, it is critical to isolate its effect on our performance. To do so, we provide an ablation study for these methods applied towards quantizing RoBERTa fine-tuned for MNLI in Table 3. While the Baseline PG k-means still exhibits consistent improvements on typical iPQ with Quant-Noise, the addition of Dense Weights Consolidation for superior initialization (Full PG k-means) noticeably improves on our proposed baseline, nearly doubling the accuracy increase from comparable compression configurations for IPQ with Quant-noise.

## 5.3 EMPTY CLUSTER RESOLUTION VIA PG k-MEANS

To demonstrate the capability of our proposed method in terms of resolving empty clusters, we gather similar statistics to our brief analysis of typical iPQ with Quant-Noise (Section 3, Table 1) and compile them in Table 4 and Table 5. Across all relevant metrics, empty clusters are extremely reduced compared to typical iPQ with Quant-Noise, in the worst case boasting around a 20x reduction in the proportion of layers with empty clusters and around a 100x reduction for the average number of empty clusters in the most problematic layers.

## 5.4 EFFICIENCY OF EMPTY CLUSTER RESOLUTION

Comparing typical iPQ with Quant-Noise's mixed heuristic and Partitioning-Guided Cluster Fine-tuning, we find that in the best case for iPQ with Quant-Noise requires 40 or more iterations of their heuristic to completely resolve empty clusters. In contrast, Partitioning-Guided Cluster Fine-tuning

Table 4: Percentages of layers with empty clusters (lower is better) for RoBERTa quantized with PG k-means and fine-tuned for MNLI, RTE, and QNLI. Compression ratios are on the left and proportions of layers with empty clusters to total layers quantized are on the right. The total number of quantized layers for RoBERTa, including sub-layers, total to 73.

| Compression Ratio | iPQ with Quant-Noise | | | PG k-means | | |
|---|---|---|---|---|---|---|
| | MNLI | RTE | QNLI | MNLI | RTE | QNLI |
| 11.81 | 94.5 | 94.5 | 93.2 | 4.1 | 2.7 | 0.0 |
| 14.05 | 79.5 | 78.1 | 78.1 | 2.7 | 4.1 | 0.0 |
| 15.29 | 82.2 | 76.7 | 79.5 | 0.0 | 1.4 | 2.7 |
| 15.90 | 76.7 | 79.5 | 78.1 | 0.0 | 2.7 | 0.0 |

Table 5: Average number of empty clusters (lower is better) per layer type in RoBERTa quantized with PG k-means and fine-tuned for MNLI, RTE, and QNLI. All results are derived from quantized models with compression ratios of 11.81 (left) and 15.9 (right). The total number of clusters for linear layers was 3072 and for embedding layers was 768. Direct comparisons can be made to iPQ with Quant-Noise results in Table 1.

| Compression Ratio of 11.81 | | | | Compression Ratio of 15.9 | | | |
|---|---|---|---|---|---|---|---|
| Layer Type | MNLI | RTE | QNLI | Layer Type | MNLI | RTE | QNLI |
| Embedding | 0.0 | 0.0 | 0.0 | Embedding | 0.0 | 0.0 | 0.0 |
| q_proj | 0.0 | 0.0 | 0.0 | q_proj | 0.0 | 0.0 | 0.0 |
| k_proj | 0.7 | 0.2 | 0.0 | k_proj | 0.0 | 0.0 | 0.0 |
| v_proj | 0.0 | 0.0 | 0.0 | v_proj | 0.0 | 0.0 | 0.0 |
| out_proj | 0.0 | 0.0 | 0.0 | out_proj | 0.0 | 0.3 | 0.0 |
| FC1 | 0.3 | 0.2 | 0.0 | FC1 | 0.0 | 0.0 | 0.0 |
| FC2 | 0.0 | 0.0 | 0.0 | FC2 | 0.0 | 0.1 | 0.0 |

requires 5 to 10 iterations on average for such cases, but its iterations are more computationally expensive. To characterize efficiency, we analyze average run-times for both methods in our evaluation environment and find that in spite of more expensive iterations, Partitioning-Guided Cluster Fine-tuning exhibits around a 3.8x speedup at worst for empty cluster resolution while on average requiring 8x fewer iterations.

# 6 CONCLUSION

In this paper, we presented partitioning-guided k-means as a competitive quantization methodology targeting extreme model compression. We compared this methodology to iPQ with Quant-Noise, the state-of-the-art scheme for quantizaion aware training and post-processing quantization for many NLP tasks and demonstrated consistently superior results for several tasks on the GLUE benchmark, producing accuracy increases of up to 2.4% for MNLI, up to 12% for RTE, and consistent increases for QNLI. Given these results, Partitioning-Guided k-means has clearly cemented itself as a strong competitor to other options for extreme model compression. Future work will involve expanding the number of applications for which we compare guided k-means to its competitors, gathering additional data to validate this approach for encoder-decoder architectures, and validating it on more compression ratios for RoBERTa fine-tuned for tasks in the GLUE benchmark.

# 7 REPRODUCIBILITY

A significant effort was made to provide sufficient material to reproduce the results generated in this paper. For a functional iPQ with Quant-Noise implementation, we refer readers to Fairseq. Concerning our method's implementation, our codebase will be made available after acceptance of this paper. Should readers wish to replicate our method internally, they should refer to the pseudo-code in this paper's Appendix and the details provided in Section 4. Model architectures and additional relevant details can be largely found in this paper's Appendix or in Section 5

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

# A APPENDIX

## A.1 FINE-TUNING AND QUANTIZATION DETAILS

All models were fine-tuned with iPQ with Quant-Noise enabled with recommended settings as provided by (Fan et al., 2020) within Fairseq's framework, keeping in line with RoBERTa's characteristics as a 12-layer model with an embedding size of 768 and an FFN hidden size of 3072. These models were fine-tuned with Adam with weight decay as an optimizer, defining $\beta_1$ and $\beta_2$ as 0.9 and 0.98, respectively, with an $\epsilon$ of 1e-6. A polynomial decay-based learning rate was applied. Dropouts were specified by LayerDrop and set to a value of 0.2. Precision for these models, by default, was 16-bit floating point. All models were evaluated via the validation split for corpora MNLI, RTE, and QNLI. All models were fine-tuned, quantized, and evaluated on four Tesla V100 SXM3s.

Compression settings were kept consistent across ratios. 768 embedding layer centroids were allocated and 3072 linear layer centroids were allocated. The quantization block sizes for product quantization of each compression ratio are shown in Table 6.

Table 6: Quantization block sizes for four compression ratios

| Compression Ratio | Block Sizes for Product Quantization | | | |
|---|---|---|---|---|
| Compr. | Linear.fc | Linear.attn | Linear.emb | Embedding.emb |
| 11.81 | 4 | 4 | 4 | 16 |
| 14.05 | 8 | 4 | 4 | 8 |
| 15.29 | 8 | 4 | 4 | 16 |
| 15.90 | 8 | 16 | 4 | 8 |

## A.2 ANECDOTAL NOTES RELATED TO OTHER TARGET APPLICATIONS AND EFFICIENCY

Simultaneous speech-to-text translation (SimulST) (Ma et al., 2020) was briefly explored as an application to assess the viability of iPQ with Quant-Noise. It was quickly observed that degenerate solutions were very common, with nearly 70% of total clusters being empty in the absolute worst case and around 48.8% in more typical cases for iPQ with Quant-Noise. We leave it to future work to explore improvements in this area.

Regarding the efficiency of our method aside from the empty cluster resolution results that were provided in the main body of this paper, there is no additional overhead in terms of test-time efficiency. This is because our method is identical to iPQ with Quant-Noise during inference. Additionally, basic k-means clustering behavior beyond pre-assignment strategies and empty cluster resolution is likewise identical, resulting in no changes to efficiency from that perspective.

## A.3 RELEVANT LICENSING INFORMATION

Fairseq (Ott et al., 2019) and any pre-trained models made available through it are MIT-licensed.

## A.4 ADDITIONAL VISUAL AIDS

A handful of additional visual aids were constructed to aid readers, but were removed due to a lack of space and redundancy with illustrations already provided within this paper. We provide them below to enable readers to engage further with this material, should they choose to do so. Figure 3 is an expansion upon what is demonstrated in Figure 1, showcasing some additional steps. Figure 4 provides an illustration of Partitioning Cluster Fine-tuning that we felt was unnecessary in the main body of this paper. Figure 5 provides an expansion upon Figure 2, showing an alternate view of its functionality and completing the demonstration of the replacement of dense clusters. As shown in Figure 6, compared with the baseline PG k-means in Figure 4, applying the optional *Partitioning-Guided Cluster Fine-tuning* step to PG k-means tends to generate the centroid distribution more faithfully to the weight distribution.

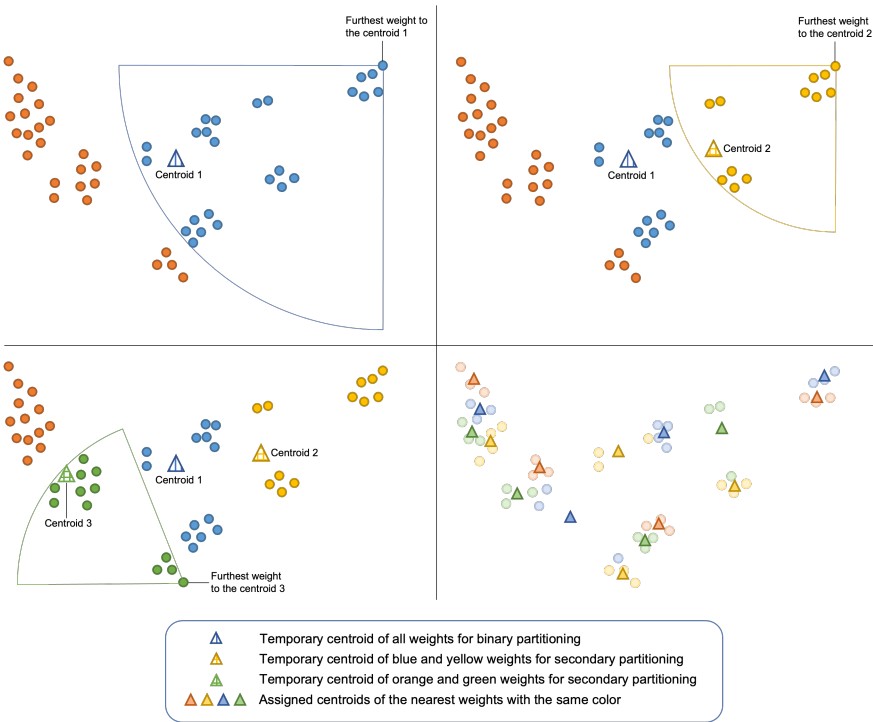

Figure 3: Illustration of *Partitioning-Guided Pre-assignment* across two partitioning time-steps when applied to a synthetic distribution. Tentative clustering is decided via $n$-dimensional, spherical partitions centered on the furthest point within the cluster of a given tentative centroid. The radius of the spherical partition targets a dynamically determined number of weights that would be assigned to the new clusters.

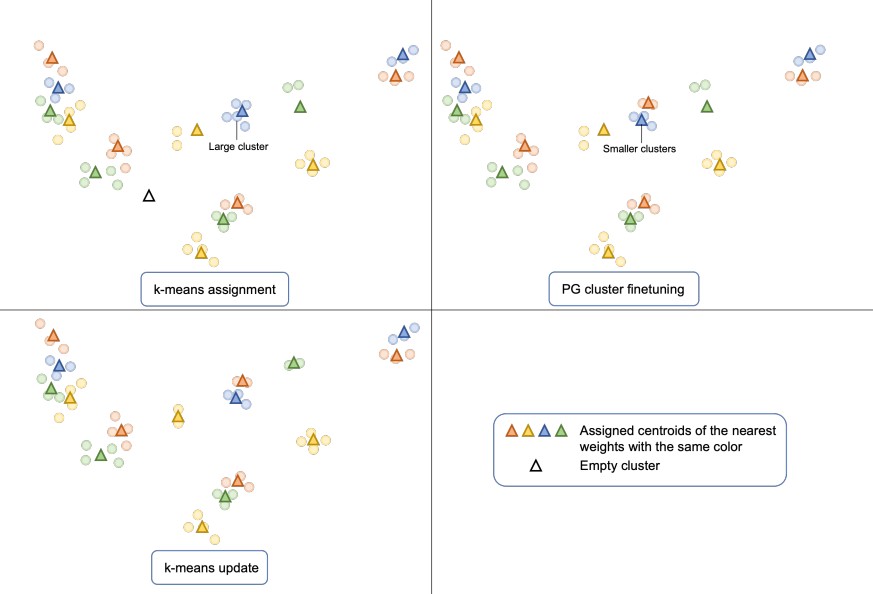

Figure 4: Illustration of *Partitioning-Guided Cluster Fine-tuning* during empty cluster resolution. For each k-means iteration, to resolve empty clusters after the k-means assignment step, *Partitioning-Guided Cluster Fine-tuning* splits large clusters into multiple smaller clusters.

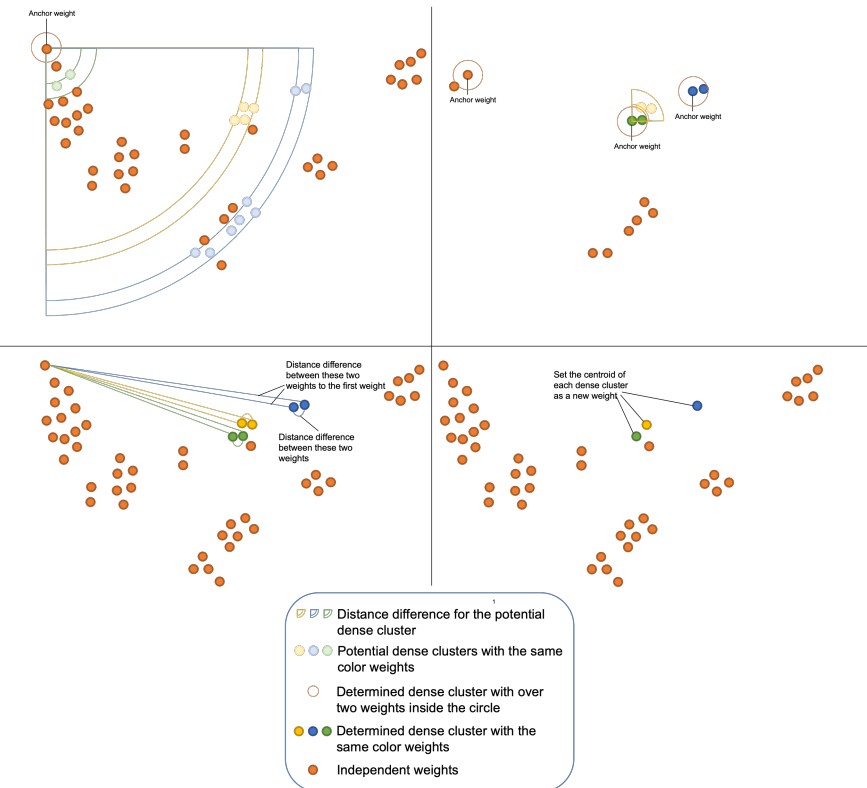

Figure 5: Illustration of *Dense Weights Consolidation* when applied to a synthetic distribution. Dense clusters are identified via Euclidean distance-based criteria. Upon dense clusters being identified, they are replaced by a centroid representing that dense cluster and treated as a normal, singular weight for later clustering steps.

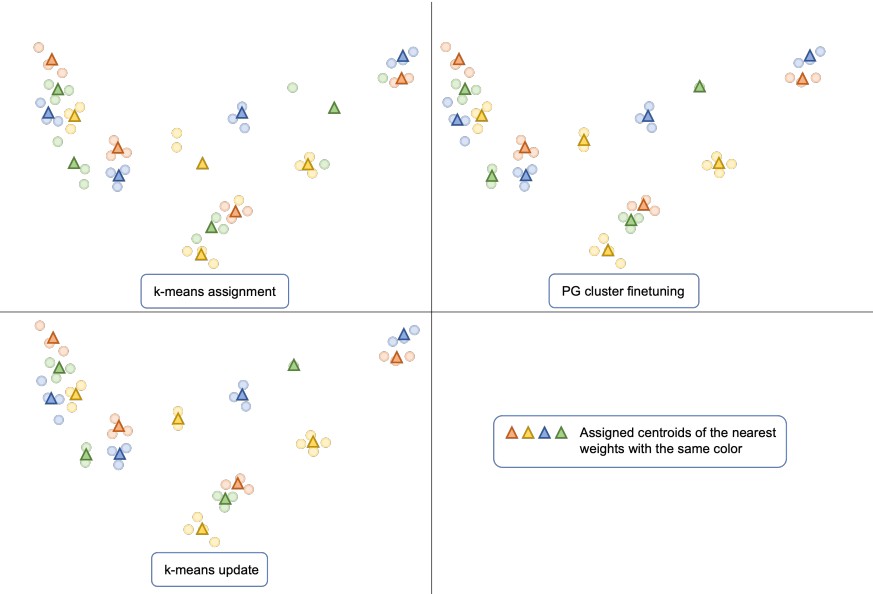

Figure 6: Illustration of complete *PG k-means* method during k-means iterations. With the optional *Dense Weights Consolidation* step, the number of weights was reduced from 50 to 47, improving our method's ability to represent isolated, small clusters while decreasing the probability of empty clusters.

### A.5 PSEUDOCODE

The pseudocode for the procedures and sub-procedures of the *Partitioning-Guided Pre-assignment*, *Partitioning-Guided Cluster Fine-tuning*, and *Dense Weights Consolidation* algorithms are defined below.

Let us denote $\mathbf{W} \in \mathbf{R}^{n \times b}$ as the weight matrix before quantizing, where $n$ is the number of weights, and $b$ is the block size of the product quantization. Alternative notation is provided in our pseudocode.

---

**Algorithm 1** Partitioning-Guided Pre-assignment

---

**Input**: Weight Matrix $W$, Centoid Matrix $C$, Average Cluster Size for each Centroid $S_{avg}$, If Reverse Last Centroid $B_{rl}$
**Output**: Centoid Matrix $C$

1: **procedure** CENTROIDPARTITIONING($W$, $C$, $S_{avg}$, $B_{rl}$)
2:     $B_{rl}$ decide if generate the last centroid or not
3:     **return** when achieved the last index of $C$ or $W$ is empty
4:     $c_w \leftarrow$ the centroid of $(W)$
5:     $n_w \leftarrow$ the number of weights in $(W)$
6:     $C \leftarrow c_w$ when $n_w \leq S_{avg} + 1$, the index of $C$ add 1, then **return**
7:     $M_c \leftarrow$ the sorted Euclidean distance map from $W$ to $C$
8:     $W_f \leftarrow$ the weight with the furthest distance to $C$ in $M_c$
9:     $M_f \leftarrow$ the sorted Euclidean distance map from $W$ to $W_f$
10:     $n_h \leftarrow$ the closest integral multiple of $S_{avg}$ to the half number of weights
11:     CENTROIDPARTITIONING(the first $n_h$ weights in $M_f$, $C$, $S_{avg}$, $B_{rl}$)
12:     CENTROIDPARTITIONING(the rest weights in $M_f$, $C$, $S_{avg}$, $B_{rl}$)
13: **end procedure**

---

---

**Algorithm 2** Partitioning-Guided Cluster Fine-tuning

---

**Input**: Weight Matrix $W$, Centoid Matrix $C$, Average Cluster Size for each Centroid $S_{avg}$
**Output**: Centoid Matrix $C$

 1: **procedure** CLUSTERFINETUNING($W$, $C$, $S_c$)
 2:     $C_e \leftarrow$ centroids with empty clusters in $C$ from the assignment
 3:     **while** $C_e$ is not empty **do**
 4:         **break** early when the number of empty clusters stops decreasing in a limited number
 5:         $C_{ra} \leftarrow C_e$                    ▷ $C_{ra}$ denotes centroids needed to be reassigned
 6:         $M_c \leftarrow$ the sorted centroid map based on the cluster size
 7:         **for** centroid $c$ in $M_c$ **do**
 8:             **if** cluser_size($c$) $\leq S_c$ **then break**                    ▷ Get the number of large clusters
 9:             **end if**
10:             $C_{ra}$.append($c$)
11:             $n_w \leftarrow n_w+$ weight_num($c$)
12:         **end for**
13:         $S_{avg} \leftarrow Max(n_w/\ \text{num}(C_{ra})\,, 1)$         ▷ Average cluster size for reassigned weights
14:         **for** centroid $c_{lc}$ of large cluster in $M_c$ **do**
15:             $W_c \leftarrow$ the weights for $c_{lc}$ in the assignment
16:             $n_{lc} \leftarrow$ weight_num($c_{lc}$)
17:             $S_{scl} \leftarrow Max(n_{lc}/\sqrt{n_{lc}/S_{avg}}, S_{avg})$        ▷ $S_{scl}$ denotes scaling sub-cluster size for splitting the large cluster
18:             CENTROIDPARTITIONING($W_c$, $C$, $S_{scl}$, True)   ▷ Reserve the last centroid $c_{last}$ for the later calculation
19:         **end for**
20:         $c_{last} \leftarrow$ the centroid of all rest weights needed to be reassigned
21:         $C$.append($c_{last}$)
22:         Recalculate empty clusters $C_e$ by updating the assignment.
23:     **end while**
24: **end procedure**

---

**Algorithm 3** Dense Weights Consolidation

---

**Input**: Original Weight Matrix $W$, Finetunable Value $\varepsilon$, Centroid Number $n_c$
**Output**: Consolidated Weight Matrix $W_c$

 1: **while** True **do**
 2:     potential dense clusters $C_{pd}$, independent weights $IW$ $\leftarrow$ IDENTIFYPOTEN-TIALDENSECLUSTER($\varepsilon$, $W$)
 3:     GENERATEDENSECLUSTERS($\varepsilon$, 0, $W$, $C_{pd}$, $C_{dd}$, $IW$)
 4:     $n_{cw} \leftarrow$ num(determined dense clusters $C_{dd}$) + num($IW$)
 5:     **if** $n_{cw} < n_c \times c_{mc}$ **then**
 6:         $\varepsilon \leftarrow \varepsilon \times c_{sd}$
 7:         **continue**
 8:     **else**
 9:         $W_c$.append(centroid for each dense cluster in $C_d$)
10:         $W_c$.append($IW$)
11:     **end if**
12: **end while**
13: **return** $W_c$

---

---

**Algorithm 4** Recursively Generate Dense Clusters

---

**Input**: Finetunable Value $\varepsilon$, Anchor Weight Index $I_a$, Weight Matrix $W$, Potential Dense Clusters $C_{pd}$, Determined Dense Clusters $C_{dd}$, Independent Weights $IW$
**Output**: Determined Dense Clusters $C_{dd}$, Independent Weights $IW$

1: **procedure** GENERATEDENSECLUSTERS($\varepsilon$, $I_a$, $W$, $C_{pd}$, $C_{dd}$, $IW$)
2:     $\triangleright$ A dense cluster is determined by if the anchor weight is in the first potential dense cluster
3:     **if then**$I_a$ in $C_{pd}[0]$
4:         $C_{dd}$.append($C_{pd}[0]$), skip the first potential dense cluster in the following loop
5:     **end if**
6:     **for** $c_p$ in $C_{pd}$ **do**
7:         $C_{subpd}, IW_{sub} \leftarrow$ IDENTIFYPOTENTIALDENSECLUSTER($\varepsilon$, weights in $c_p$)
8:         $IW$.append($IW_{sub}$)
9:         **if** $C_{subpd}$ is not empty **then**
10:             GENERATEDENSECLUSTERS($\varepsilon$, $c_p[0]$, $W$, $C_{subpd}$, $C_{dd}$, $IW$)
11:         **end if**
12:     **end for**
13: **end procedure**

---

**Algorithm 5** Identify Potential Dense Clusters

---

**Input**: Finetunable Value $\varepsilon$, Weight Matrix $W$
**Output**: Potential Dense Clusters $C_{pd}$, Independent Weights $IW$

1: **function** IDENTIFYPOTENTIALDENSECLUSTER($\varepsilon$, $W$)
2:     $M_w \leftarrow$ the sorted Euclidean distance map from $W$ to $W[0]$
3:     $s \leftarrow 0$
4:     **for** index $i$ of distance in $M_w$ **do**
5:         **if** $M_w[i] - M_w[s] > \varepsilon$ **then**
6:             **if** $i - s > 1$ **then**
7:                 $C_{pd}$.append($M_w[s:i]$)       $\triangleright$ Append weights in $M_w$ between indices $s$ and $i$
8:             **else**
9:                 $IW$.append($M_w[s]$)           $\triangleright$ Append the weight in $M_w$ on index $s$
10:             **end if**
11:             $s \leftarrow i$
12:         **end if**
13:     **end for**
14:     **return** $C_{pd}$, $IW$
15: **end function**

---

