# OpenReview forum: "Partitioning-Guided K-Means: Extreme Empty Cluster Resolution for Extreme Model Compression"
_ICLR.cc/2024/Conference — ICLR 2024 Conference Withdrawn Submission_

### Official Review · Reviewer_uYXA · 2023-10-18

**Soundness:** 3 good
**Presentation:** 2 fair
**Contribution:** 2 fair
**Rating:** 3
**Confidence:** 4

**Summary:**

The paper proposes Partitioning-Guided k-means (PG k-means), a novel k-means algorithm for quantizing neural network weights. It aims to improve Iterative Product Quantization (iPQ) with Quant-Noise, which suffers from empty clusters that degrade accuracy.

PG k-means has 3 main components: Partitioning-guided pre-assignment - Minimizes initial empty clusters by recursively partitioning weights.
Partitioning-guided cluster fine-tuning - Resolves empty clusters during k-means by cautiously splitting large clusters. Dense weights consolidation (optional) - Consolidates dense clusters to prevent erroneous separation. PG k-means consistently outperforms iPQ with Quant-Noise for extreme compression of RoBERTa on GLUE benchmarks, demonstrating its viability for quantization-aware training. The proposed techniques significantly reduce empty clusters and improve accuracy.

**Strengths:**

PG k-means consistently outperforms iPQ with Quant-Noise for extreme compression of RoBERTa on GLUE benchmarks, demonstrating its viability for quantization-aware training. The proposed techniques significantly reduce empty clusters and improve accuracy.

**Weaknesses:**

At the beginning of the introduction, the authors claimed their object is to compress the large language models (LLMs). However, I have not seen any evidences showing that their method is beneficial for LLM compression. Specifically, all the explorations and experiments are conducted with RoBERTa networks. Those BERT-architecture models only have 1 billion parameters, which can not be called LLM for the recent community. And the architecture of RoBERTa is hugely different than the architecture of those real LLMs such as GPT, OPT, and LLAMA.

The scope is limited, the authors choose to modify or improve Iterative Product Quantization (iPQ) with Quant-Noise (Fan et al.,
2020), as it is the state-of-the-art in this area. In the area of network (non-uniformed) quantization, there are many works can outperform Iterative Product Quantization (iPQ) with Quant-Noise. The foundation of developing their method based on iPQ with Quant-Noise is not solid.

**Questions:**

Refer to weaknesses.

---

### Official Review · Reviewer_rfms · 2023-10-30

**Soundness:** 2 fair
**Presentation:** 3 good
**Contribution:** 2 fair
**Rating:** 5
**Confidence:** 4

**Summary:**

This paper presents Partitioning-Guided k-means (PG k-means) as a new approach to address the issue of inference quality degradation caused by empty clusters in Iterative Product Quantization (iPQ) with Quant-Noise. The proposed PG k-means method comprises three key components to resolve the prevalent empty cluster problem. The PG k-means offers practical solutions to minimize empty clusters, leverage improved empty cluster resolution, and enhance overall accuracy for extreme model compression in language modeling tasks.

**Strengths:**

The proposed PG k-means method reduces the number of empty clusters in iPQ with Quant-Noise, decreases the iterations required for empty cluster resolution, and improves the overall model accuracy by up to 12% when applied to RoBERTa on various tasks in the GLUE benchmark. The effectiveness of PG k-means is compared to iPQ with Quant-Noise, revealing consistently superior results on tasks such as MNLI, RTE, and QNLI. The findings suggest that PG k-means is a solution for extreme model compression.

**Weaknesses:**

I have some concerns as follows.

This paper falls in the scope of PQ-based model compression [1-3]. Previous work has typically focused on quantizing model parameters for compression purposes [4-5], it is unsurprising that optimizing the k-means clustering inside can improve model performance.

[1] Product quantization for nearest neighbor search. TPAMI 2010.
[2] VQ-GNN: A Universal Framework to Scale up Graph Neural Networks using Vector Quantization. NeurIPS 2021.
[3] Quantized convolutional neural networks for mobile devices. CVPR 2016.
[4] Dynamic Dual Gating Neural Networks. ICCV 2021.
[5] FPGM-Filter Pruning via Geometric Median for Deep Convolutional Neural Networks Acceleration. CVPR 2019.

**Questions:**

Please see the weakness part.

---

### Official Review · Reviewer_5ufE · 2023-10-30

**Soundness:** 1 poor
**Presentation:** 3 good
**Contribution:** 2 fair
**Rating:** 3
**Confidence:** 5

**Summary:**

This paper focuses on the extreme model compression with product quantization. The authors first reveal the empty cluster issue in Iterative Product Quantization (iPQ) method proposed in 2020, and they posit that the presence of empty clusters often leads to noteworthy losses in inference quality. To solve this problem, Partitioning-Guided k-means (PG k-means) is proposed with three contributions:

- a pre-assignment strategy that ensures non-empty initial centroids.

- a partitioning method of populous clusters into new sub-clusters when empty clusters arise during k-means iterations.

- an optional optimization step that consolidates dense clusters of weights to ensure that they map to a single centroid after quantization completes.

**Strengths:**

The proposed method about solving empty cluster problem is reasonable and is easy to follow.
Detailed algorithm implementation is given.

**Weaknesses:**

I cannot agree with the authors about the empty cluster problem to cause the loss in model quality. In popular k-means implementation, there are mechanisms to make sure that the clusters are non-empty, such as the k-means++ initialization and the reassignment method used in scikit-learn KMeans. I believe the k-means method is not correctly used if lots of clusters are empty.

If the iPQ with Quant-Noise can resolve empty clusters with more iterations, then the proposed method only speedup the k-means optimization step. From this viewpoint, the necessity of the proposed method is questionable.

The proposed method is validated on a single RoBERTa model. The three tasks (MNLI, RTE, QNLI) of GLUE are also not enough.

The baseline results of iPQ with Quant-Noise seems to be much lower than reported in the original paper. In iPQ, the accuracy on MNLI is 83.6 with 12.6x compression and 82.5 with 34.3x compression, which outperforms the proposed PG k-means method.

The proposed method should be validated on other models such as EfficientNet-B3 to fairly compare with iPQ. Comparison with other quantization methods, such as binary/ternary quantization methods, should be discussed.

1. https://github.com/scikit-learn/scikit-learn/blob/main/sklearn/cluster/_kmeans.py

**Questions:**

When calculating the model size of the original model, what data format is used (Fp32 or fp16)?
What are the percentages when Partitioning-Guided Pre-assignment method is used without Partitioning-Guided Cluster Fine-tuning?